# Transcriptome Profiling Provides New Insights into the Molecular Mechanism Underlying the Sensitivity of Cotton Varieties to Mepiquat Chloride

**DOI:** 10.3390/ijms23095043

**Published:** 2022-05-02

**Authors:** Zhijun Wang, Yanjun Li, Qianhao Zhu, Liwen Tian, Feng Liu, Xinyu Zhang, Jie Sun

**Affiliations:** 1The Key Laboratory of Oasis Eco-agriculture, Agriculture College, Shihezi University, Beisi Road, Shihezi 832003, China; wzjshihezi@163.com (Z.W.); lyj20022002@sina.com.cn (Y.L.); liufeng@shzu.edu.cn (F.L.); 2Biotechnology Research Institute, Xinjiang Academy of Agriculture and Reclamation Science, Shihezi 832000, China; 3CSIRO Agriculture and Food, GPO Box 1700, Canberra 2601, Australia; qianhao.zhu@csiro.au; 4Cash Crop Research Institute, Xinjiang Agricultural Academy, Urumqi 830091, China; jzskg@sina.cn

**Keywords:** mepiquat chloride (MC), RNA sequencing (RNA-seq), *Gossypium hirsutum* L., phytohormones, transcription factor

## Abstract

Mepiquat chloride (MC) is a plant growth regulator widely used in cotton production to control vegetative overgrowth of cotton plants to achieve ideal plant architecture required for high yielding. Cotton varieties respond differently to MC application, but there is little information about the molecular mechanisms underlying the varietal difference. In this study, comparative transcriptome analysis was conducted by using two Upland cotton varieties with different sensitivity (XLZ74, insensitive; SD1068, sensitive) to MC treatment, aiming to understand the molecular mechanisms responsible for varietal difference of MC sensitivity. RNA-seq data were generated from the two varieties treated with MC or water at three time points, 1, 3 and 6 days post-spray (dps). Genes differentially expressed between the MC and mock treatments of XLZ74 (6252) and SD1068 (6163) were subjected to Gene Ontology (GO) and Kyoto Encyclopedia of Genes and Genomes (KEGG) analyses to compare the enriched GO terms and KEGG pathways between the two varieties. Signal transduction of phytohormones, biosynthesis of gibberellins (GAs) and brassinosteroids (BRs) and profiles of transcription factors (TFs) seemed to be differentially affected by MC in the two varieties. The transcriptomic results were further consolidated with the content changes of phytohormones in young stem. Several GA catabolic genes, *GA2ox*, were highly induced by MC in both varieties especially in SD1068, consistent with a more significant decrease in GA_4_ in SD1068. Several *AUX/IAA* and *SAUR* genes and *CKX* genes were induced by MC in both varieties, but with a more profound effect observed in SD1068 that showed a significant reduction in indole-3-acetic acid (IAA) and a significant increase in cytokinin (CTK) at 6 days post-spray (dps). BR biosynthesis-related genes were downregulated in SD1068, but not in XLZ74. Additionally, more downregulated TFs were observed in MC-treated SD1068 than in MC-treated XLZ74, and the two varieties had very different profiles of genes involved in starch and sucrose metabolism, with those of SD1068 and XLZ74 being downregulated and upregulated by MC treatment, respectively. Together, these results indicate that although the same or similar biological pathways are affected by MC treatment in cotton varieties showing different MC sensitivity, the extent of effect is variable, leading to their different phenotypic outcomes. How the quantitative effect of MC on the biological processes associated with growth retardation is regulated is still an open question.

## 1. Introduction

Cotton (*Gossypium hirusutum* L.) is one of the most important fiber crops cultivated in more than seventy countries. Cotton is a perennial shrub with indeterminate growth habit. Improper field management will lead to cotton with excessive vegetative growth, resulting in self-shading, fruit abortion, delayed maturity and low productivity [1,2]. Mepiquat chloride (MC) is a kind of plant growth regulator used to control plant growth by foliar spray [3]. MC has been regularly used in cotton production to enhance yield outcome by shaping plant architecture through restricting internode elongation, reducing plant height and promoting boll setting and development [4,5]. MC has also been shown to enhance lateral root formation in cotton [6,7]. Because of these beneficial impacts of MC on cotton plant growth and development, MC application has become one of the key agronomic measures of commercial cotton production [1].

Gibberellic acids (GAs) are plant growth-promoting hormones that play important roles in regulating stem and internode elongation by affecting cell division and expansion [8,9]. MC controls plant growth by inhibition of GA biosynthesis [10] and interference of the homeostasis of other phytohormones, including auxin, brassinosteroid (BR), cytokinin (CTK), abscisic acid (ABA) and ethylene (ETH) [7,11]. As a result, MC modulates the metabolism and distribution of carbohydrates, enzymes and other organic molecules essential for plant growth and development [12,13]. These studies revealed the physiological effects of MC on plant morphogenesis [11,12,13]. Using transcriptome profiling, some of these studies also explored the potential molecular mechanisms underlying the physiological impacts of MC-mediated inhibition of internode elongation and promotion of root formation [7,11].

Cotton varieties differ significantly in their response to MC treatment. Generally, they can be divided into insensitive, intermediate and sensitive three categories [14,15]. MC-sensitive cotton varieties need only 1 to 2 times of foliar sprays low concentration of MC (15–30 g/hm^2^) during their growth season [16]. MC-insensitive cotton varieties need 3–4 times of spray of relatively high concentration of MC (75 g/hm^2^) in order to balance their vegetative and reproductive growth. Despite the observation of varietal difference in MC sensitivity, no study has investigated the molecular mechanisms responsible for the variable responses of cotton varieties to MC treatment.

To address this issue, in this study, we did a comparative transcriptomic analysis using two cotton varieties, Xinluzao 74 (XLZ74) and Shida1068 (SD1068) that are insensitive and sensitive to MC treatment, respectively. The transcriptomes of young stem of the two varieties at different time points following MC or water treatment were examined and used to identify genes differentially expressed between the treatments. GO and KEGG analyses were used to characterize the MC-responsive genes and pathways. The results provide new insights into the molecular mechanism underlying MC-mediated growth retardation and varietial difference in response to MC treatment, which will guide effective use of MC and broaden the application scope of MC in crop production. 

## 2. Results

### 2.1. The Sensitivity of Cotton Varieties to MC

In order to compare the sensitivity of SD1068 and XLZ74 to MC, seedlings of the two varieties were treated three times with foliar spraying of 40, 80 and 120 mg/L MC. Compared with the control plants (water treatment), the plant height of XLZ74 treated by 40, 80 and 120 mg/L MC decreased by 19.32, 23.38 and 33.74%, respectively. For SD1068, the plant height of the corresponding treatments decreased by 27.2, 47.10 and 48.05%, respectively (Figure 1a). In order to further compare the sensitivity of the two varieties with MC, plant height and the length of the second and third internodes were measured for the seedlings treated with 80 mg/L MC at 2, 4, 6, 8 and 10 dps. Compared with the control plants, the plant height of SD1068 increased by 3.01% at 2 dps, and then decreased by 1.67, 4.98, 12.54 and 13.60% at 4, 6, 8, and 10 dps, respectively. The length of the second internode was reduced by 8.59, 14.54, 7.49, 13.75 and 19.63%, and the third internode by 1.69, 2.70, 13.01, 13.53 and 14.06% at 2, 4, 6, 8 and 10 dps, respectively. The length of the second internode was inhibited more severely than that of the third internode after MC treatment. A significant reduction in plant height was observed after 8 dps and a significant reduction in the internode length was observed after 6 dps.

For XLZ74, the plant height and the length of the second and third internodes were not significantly inhibited. The plant height decreased by 1.54, 1.53, 0.75, 0.44 and 0.51% at 2, 4, 6, 8, and 10 dps, respectively. The length of the second internode was reduced by 6.23, 4.82, 5.07, 5.73 and 6.39%, and the third internode by 0.24, 0.31, 1.29, 1.31 and 1.33% at 2, 4, 6, 8, and 10 dps, respectively (Figure 1b). There was no significant difference between control and MC-treated plants.

### 2.2. An Overview of the RNA-Seq Data

To determine the transcriptome profile of cotton in response to MC, RNA-seq was performed for XLZ74 and SD1068 at 1, 3 and 6 dps with MC and water treatments. In total, we generated 36 RNA-seq datasets, of which 18 from XLZ74 following MC or water treatment (XD-1, XD-3, XD-6, XW-1, XW-3 and XW-6, each with three replicates), 18 from SD1068 (SD-1, SD-3, SD-6, SW-1, SW-3 and SW-6, each with three replicates).

After RNA sequencing, the quality of the data was assessed. An overview of the sequencing results was outlined in Appendix A. After filtering the low-quality reads, a total of 284.11 Gb high-quality clean data were generated. All libraries had high Q30 percentage of approximately 93.53% and a constant GC content (GC_pct) of approximately 43.38%. The proportion of clean data mapped to the reference genome (proper_map) varied from 88.32 to 90.42%, and that of the uniquely mapped reads (unique_map) varied from 91.55 to 92.34%. The data of the 36 libraries also indicated a good level of reproducibility, providing a solid foundation for identifying key genes participating in regulating cotton development in response to MC treatment (Appendix A). The reliability of the RNA-seq results was further confirmed by qRT-PCR using 9 randomly selected genes (Appendix A, Figure 2), as demonstrated by the highly consistent results between RNA-seq and qPCR.

### 2.3. Analyses of Differentially Expressed Genes (DEGs)

To determine the gene expression changes in the young stem from the two cotton varieties following MC treatment, screening of DEGs by comparing water and MC treatments at the same time point was conducted. Compared with water treatments, the gene expression of both varieties changed significantly after MC treatment. For the MC insensitive variety XLZ74 (Figure 3a), 798 (200 upregulated and 598 downregulated), 3719 (2335 upregulated and 1384 downregulated) and 3007 (1708 upregulated and 1299 downregulated) DEGs were identified at 1, 3, and 6 dps, respectively. For the MC-sensitive variety SD1068 (Figure 3a), 1247 (581 upregulated and 666 downregulated), 2234 (1209 upregulated and 1025 downregulated) and 3406 (1628 upregulated and 1778 downregulated) DEGs were identified at 1, 3, and 6 dps, respectively. The number of DEGs (1247) at 1 dps in SD1068 was more than that in XLZ74 (798), indicating that the response of SD1068 to MC was more sensitive than that of XLZ74. By contrast, XLZ74 had more upregulated genes than SD1068 at 3 and 6 dps. As shown in the Venn diagrams, there were 6252 DEGs identified in XLZ74, of which 68 DEGs (1.09%) were found at all three time-points, whilst 5048 DEGs (80.74%) were found at only one time point, including 545 DEGs at 1 dps, 2595 at 3 dps, and 1908 at 6 dps (Figure 3b). About 105 DEGs were shared between 1 and 3 dps, 951 DEGs between 3 and 6 dps, and 80 DEGs between 1 and 6 dps. There were 6163 DEGs identified in SD1068, of which 58 (0.94%) were identified at all three time-points, whilst 5497 DEGs (89.19%) were identified at only one time point, including 869 DEGs at 1 dps, 1728 at 3 dps, and 2900 at 6 dps (Figure 3c). About 160 DEGs were shared between 1 and 3 dps, 288 DEGs between 3 and 6 dps, and 160 DEGs between 1 and 6 dps.

### 2.4. GO and KEGG Analyses

Gene Ontology enrichment analysis was applied to all the DEGs to identify active biological processes involved in cotton’s response to MC treatment. The top ten GO terms enriched in SD1068 were significantly different from those in XLZ74 (Figure 4a). For SD1068, most DEGs were associated with hormones, including ‘response to hormones’, ‘cytokinin metabolic process’, ‘regulation of hormone levels’, ‘cellular hormone metabolic process’ and ‘hormone metabolic process’. By contrast, hormone-related DEGs were not on the list of the top ten GO terms in XLZ74. The notable GO terms in XLZ74 were associated with “sulfate transport” and “sulfate compound transport”. GO enrichment analysis suggested that the molecular mechanism underlying the varietal difference in response to MC may be different.

KEGG enrichment analysis was performed to explore the biological pathways of DEGs responsive to MC in the two varieties. The top 20 KEGG pathways enriched in each variety were shown in Figure 4b. Several pathways related to hormone metabolism (*p* < 0.001) were the same in the two varieties, including ‘plant hormone signal transduction’, ‘zeatin biosynthesis’ and ‘diterpenoid biosynthesis’ pathway. Pathways unique to XLZ74, included ‘phenylopropanoid biosynthesis’, ‘cutin, suberine and wax biosynthesis’ and ‘tropane, piperidine and pyridine alkaloid biosynthesis’, and pathways unique to SD1068 included ‘brassinosteroid biosynthesis’, ‘photosynthesis-antenna proteins’, ‘carotenoid biosynthesis’ and ‘nitrogen metabolism’ (Figure 4b). In order to further find the response differences of the two cultivars, KEGG pathway analysis was also performed for the up and downregulated DEGs of the two vareties at each time point (Appendix A). It was notable that DEGs related to ‘brassinosteroid biosynthesis’ and ‘starch and sucrose metabolism’ were enriched (*p* < 0.001) at different time point in the two varieties, and showed different expression profile. The ‘brassinosteroid biosynthesis’ pathway was downregulated at 6 dps in SD1068, whereas upregulated at 3 dps in XLZ74. The ‘starch and sucrose metabolism’ pathway was downregulated at 6 dps in SD1068, whereas it was upregulated at 6 dps in XLZ74.

### 2.5. DEGs Related to Hormone Metabolism and Signaling Pathways

The KEGG pathway analysis suggested that several DEGs related to metabolism and signaling of multiple hormones, including GAs, IAA, CTK and BR, were differentially expressed following MC treatment in both SD1068 and XLZ74.

The expression of 17 and 16 genes related to GA metabolism and signaling significantly changed in XLZ74 and SD1068, respectively. *CPS*, *KAO1* and *GA3ox* play important roles in GAs biosynthesis, whereas *GA2ox* deactivates active GAs (Figure 5a). In XLZ74 (Figure 5b), 8 genes were found to be downregulated at 1 dps, including 4 signaling genes (*GID1B*), 3 catabolic genes (2 *GA2ox1* and 1 *CYP82G1*) and 1 biosynthetic gene (*GA3ox1*). Most GA-related genes were found to be upregulated at least at one time point (3 or 6 dps), including 5 catabolic genes (2 *GA2ox1* and 3 *GA2ox2*), 3 biosynthetic genes (1 *CPS*, 1 *KAO1* and 1 *GA3ox1*) and 3 signaling genes (*GID1B*). In SD1068 (Figure 5c), 5 genes (4 *GA2ox1* and 1 *GA20ox2*) were downregulated, and 4 genes (3 *GA2ox* and 1 *GA20ox2*) were upregulated at least at one time point (1 or 3 dps). Most showed expression change at 6 dps, including 7 downregulated genes (4 *GA2ox*, 2 *CYP82G1* and 1 *GA20ox2*) and 9 upregulated genes (6 *GA2ox* and 3 *GA20ox*). It was notable that several *GA2ox* genes showed more than 3-fold upregulation after MC treatment in both varieties. Of these genes, 4 *GA2ox* genes (*GH_D07G1398*, *GH_A07G1409*, *GH_D06G1621* and *GH_D09G0919*) were induced 3- to 6-fold by MC at least at one time piont in XLZ74, and 3 *GA2ox* genes (*GH_A05G1688*, *GH_D09G0919* and *GH_A10G0785*) were induced more than 7-fold by MC at 6 dps in SD1068.

It was found that the expression of 43 and 36 DEGs related to auxin pathway (Figure 6a) significantly changed in XLZ74 and SD1068, respectively. Auxin regulates the expression of hundreds of genes [17]. Among these genes, *Aux/IAA*, *GH3*, and *SAUR* (SMALL AUXIN UP RNA) are three large primary auxin-induced genes. In XLZ74 (Figure 6b), 4 *GH3* and 1 *IAA29* were downregulated at 1 dps, whereas 36 genes, including 4 *GH3*, 13 *IAA*, 12 *AUX* and 7 *SAUR*, were obviously upregulated at 3 dps. In SD1068 (Figure 6c), 1 *GH3.17*, 1 *IAA4*, 3 *AUX* and 3 *SAUR* were downregulated at 1 dps or 3 dps, whereas 22 genes (3 *IAA*, 10 *AUX* and 9 *SAUR*) were obviously upregulated at 6 dps.

There were 21 and 19 CTK-related DEGs identified in XLZ74 and SD1068, respectively. CKX (Cytokinin dehydrogenase) enzyme catalyzes the irreversible degradation of cytokinins [18]. *ARR* (Two-component response regulator) genes are the primary responsive genes in the CTK signaling pathway [19]. A total of 11 *CKX* and 9 *ARR* were found to be upregulated at least at one time point (3 or 6 dps) in XLZ74 (Figure 7a). Six *CKX* genes were upregulated at least at one time point in SD1068, including 3 significantly induced by MC from 1 dps (Figure 7b). LOG (Cytokinin riboside 5′-monophosphate phosphoribohydrolase) has been identified as a cytokinin-producing enzyme, which catalyzes the final step in synthesis of bioactive CTKs [20,21]. A total of 10 *LOG* genes were downregulated in SD1068 at 6 dps, whereas no *LOG* genes were found to be differentially expressed in XLZ74 (Figure 7a,b).

Six *CYP* genes involved in brassinosteroid (BR) biosynthesis were significantly downregulated after MC treatment in SD1068 at 6 dps. However, four *CYP* genes were upregulated at least at one time point (3 or 6 dps) after MC treatment in XLZ74. *BZR1* transcription factor functions as a master regulator of BR signaling [22]. One *BZR1* was found to be upregulated at 6 dps in XLZ74 (Figure 7c,d).

Taken together, KEGG analyses found that hormone metabolism and signaling pathways were the most notably enriched in both varieties, likely responsible for the inhibitory effect of MC on plant growth. The plant height and the length of the second and third internodes were obviously inhibited at 8 dps in SD1068, whereas they were not significantly inhibited in XLZ74, suggesting that the inhibitory effect of MC on SD1068 was more severe than that on XLZ74. The expression of *GA20x* showed more than 7-fold upregulation after MC treatment in SD1068, but showed only 3- to 5- fold upregulation in XLZ74. The expression increase in many auxin-induced genes (*IAA/AUX* and *SAUR*) in SD1068 was delayed (6 dps) compared to XLZ74 (3 dps). The expression of CTK degradation signaling genes (*CKX*) induced by MC from 1 dps in SD1068, but from 3 dps in XLZ74. The expression of several *ARR* genes was upregulated only in XLZ74, and *LOG* was downregulated only in SD1068. Several BR biosynthesis-related genes were found to be downregulated only in SD1068. The differences in the expression of these hormone metabolism and signaling genes between the two varieties may be partly responsible for the findings that the plant height and the internode length were degraded more in SD1068 than in XLZ74 after MC treatment.

### 2.6. The Effect of MC on the Contents of Endogenous Hormones in the Two Varieties

To investigate the effect of MC on the changes of endogenous hormones, GAs, IAAs and CTK contents in young stems of water and MC-treated plants at 6 dps were quantified. It was found that the content of some of GAs (GA_4_ and GA_15_) and IAAs (IAA-Glu and IAA-Glc) decreased, whilst CTK (tZR) increased in both varieties following MC treatment. Trp is a biosynthetic precursor of IAAs, and the content of Trp also decreased significantly. However, the content of other types of GAs, IAA and CTK did not change significantly. The content of GA_4_ was reduced by 21.6 and 47.1%, Trp by 20.12 and 65.62%, IAA-Glc by 2.4 and 71.2% and IAA-Glu by 21.0 and 30.1% in XLZ74 and SD1068, respectively (Figure 8a–e). The content of tZR was increased by 28.8 and 105% in XLZ74 and SD1068, respectively (Figure 8f). Compared to XLZ74, SD1068 showed a more significant decrease in GA_4_, Trp, IAA-Glc and Glu-Glu content, and a more significant increase in CTK content. However, a more significant decrease in GA_15_ content was found in XLZ74. The content of GA_15_ was reduced by 90 and 86.5% in XLZ74 and SD1068, respectively.

### 2.7. Differentially Expressed Transcription Factors

The expression of a number of transcription factors (TFs) was found to be changed in XLZ74 and SD1068 following MC treatment. The TFs whose expression increased at least at two time points was considered to be upregulated TFs, and decreased at least at two time points was downregulated TFs. For SD1068, the number of downregulated TFs were significantly higher than that of upregulated ones at all three time points (Figure 9a). For XLZ74, there were more downregulated TFs at 1 dps, but more upregulated TFs at 3 and 6 dps (Figure 10a). Among these TFs, *MYB*, *NAC*, *bHLH* and *WRKY* were significantly enriched at least at one time point in both varieties, suggesting that they may play important role in the MC-induced growth retardation. The number of downregulated *MYB*, *NAC*, *bHLH* and *WRKY* was greater than that of upregulated ones in SD1068 (Figure 9a–c). By contrast, less downregulated TFs were found in XLZ74 (Figure 10a–c).

To identify the relationship between TFs and hormone-related genes, we constructed a co-expression network of TFs and hormone metabolism and signaling genes (Figure 9d and Figure 10d). The correlation of co-efficient between genes was calculated separately using expression data with a threshold of 0.8 for a positive correlation and −0.8 for a negative correlation (*p* < 0.05). For SD1068, a total of 38 TFs were found to have a high degree of co-expresison relationship with 18 hormone metabolism and signaling genes (Appendix A; Figure 9d). Among the structural genes, auxin-related gene *AUX15A* linked to the largest number of TFs, including *WRKY53*, *TCP14*, *MYB20*, *NAC056*, *NAC002*, *BZIP61*, *BHLH30* and *OsC3H49*. Followed by *GA2ox8* and *SAUR50*, each linked to seven TFs. *GA2ox8* linked to *ZBTB8A*, *WRKY46*, *WRKY40*, *MYB86*, *ERF9*, *BZIP44* and *AUX22B*, and *SAUR50* linked to *TCP20*, *RAP2-7*, *MYB20*, *NAC083*, *MYB6*, *DREB3* and *BHLH30*. It was also found that *CKX7*, *GH3.17* and *IAA4* each linked to six TFs. The connected TFs included 38 TFs belonging to 11 families, of which *OsC3H49* linked to 7 structural genes (*PP2C56* and *PP2C37* (ABA-related genes), *LOG1*, *GA2ox2*, *CYP90A1*, *CKX7* and *AUX15A*). Other identified TFs included *MYB20*, *NAC083*, *DREB3*, each linked to 5 structural genes.

For XLZ74, a total of 38 TFs were found to have a high degree of co-expression relationship with 17 hormone metabolism and signaling genes (Appendix A; Figure 10d). Among the structural genes, auxin-related genes *AUX15A* linked to the largest number of TFs, including *WRKY70*, *RL6*, *PRE5*, *NAC043*, *NAC035*, *GL3*, *ERF105*, *ERF053* and *AP2*. Followed by *SAUR50*, *IAA29*, *CKX6* and *ARR4*, each linked to 8 TFs. *SAUR50* linked to *PRE5*, *PIF1*, *MYB20*, *NAC104*, *NAC021*, *MYBS3*, *DREB3* and *BHLH93*, *IAA29* linked to *NAC002*, *MYBS3*, *MYB3*, *NAC42*, *ERF053*, *DREB3*, *BHLH96* and *BHLH93*, *CKX6* linked to *RL6*, *RL3*, *PRE5*, *MYB20*, *NAC047*, *DREB3*, *BHLH93* and *BHLH63*, and *ARR4* linked to *WRKY70*, *PIF1*, *NAC071*, *NAC043*, *NAC035*, *NAC002*, *ERF105* and *AP2*. It was also found that *GA2ox1*, *AUX22D* and *GH3.6* each linked to six TFs. The connected TFs included 38 TFs belonging to 10 families, of which *DREB3* linked to 7 structural genes (*PP2C75*, *GA2ox1*, *IAA29*, *SAUR50*, *CKX6*, *GA3ox1* and *GID1B*). Followed by *PRE5*, *WRKY70* and *RL3*, each linked to five structural genes. Other identified important TFs included *MYB20*, *BHLH93* and *AP2*, each linked to four structural genes.

### 2.8. DEGs Related to Metabolism of Starch, Sucrose and Phenylpropanoid

KEGG enrichment showed that DEGs related to starch and sucrose metabolism were obviously changed in both varieties. The metabolism of starch and sucrose fuels all aspects of plant growth and development. In XLZ74, 5 starch and sucrose-related DEGs were downregulated at 1 dps, but 20 related genes were upregulated at least at one time point (3 or 6 dps). In SD1068, 20 starch and sucrose-related genes were found to be downregulated at 6 dps. Trehalose plays important roles in plant growth and stress responses and is synthesized from trehalose-6-phosphate by trehalose-6-phosphated phosphatase (TPP). Four and seven *TPP* genes differentially expressed following MC treatment in SD1068 and XLZ74, respectively, but showed different expression patterns (Figure 11a). The 4 differentially expressed *TPP* genes found in XLZ74 were upregulated at 3 and 6 dps, whereas the 7 differentially expressed *TPP* genes found in SD1068 were downregulated at 6 dps. Four *BAM* (β-amylase) genes were downregulated at 6 dps in SD1068, but no differentially expressed *BAM* was found in XLZ74.

KEGG enrichment showed that DEGs involved in phenylpropanoid metabolism were enriched in response to MC in XLZ74, but not in SD1068. The phenylpropanoid pathway provides metabolites for plant growth, which contributes to the requirement of lignin and flavonoid biosynthesis. DEGs related to lignin biosynthetic pathway, including 5 *P**ER* (peroxidase), 3 *4CL* (4-coumarate-CoA ligase 2), 2 *COMT* (caffeic acid 3-O-methyltransferase), 2 *CAD*, 1 *ALDH* (aldehyde dehydrogenase), and 1 *PAL* (phenylalanine ammonialyse), were downregulated at least at one time point by MC. Only 6 DEGs related to lignin biosynthetic pathway were upregulated, inculing 4 *PER*, 1 *CAD*1 and 1 *OMT*1. In total, 9 DEGs involved in flavonoid biosynthesis, including 4 *CHS*, 1 *CHI*, 2 *F3H* and 2 *AN3*, were identified. All these DEGs were repressed by MC at least at one time point (Figure 11b).

## 3. Discussion

Previous reports found that MC can reduce internode elongation and plant height [11,23]. In this study, the plant height, the second and third internodes of SD1068 were significantly inhibited by MC, consistent with the previous reports. However, the plant height, the second and third internodes of XLZ74 were not inhibited by MC. These observations suggest that the inhibitory effect of MC on SD1068 was more severe than that on XLZ74. In other words, SD1068 was more sensitive to MC than XLZ74. RNA-seq was conducted to find MC-responsive DEGs in XLZ74 and SD1068 at different time points (1, 3 and 6 dps). KEGG analyses revealed common MC-responsive pathways and TFs in the two varieties, as well as pathways unique to XLZ74 or SD1068. The pathways which may be responsible for the differing sensitivity of the two varieties to MC are discussed below.

### 3.1. MC Treatment Induces Significant Upregulation of GA Catabolic Genes GA2ox, Especially in SD1068

Previous reports have showed that MC suppressed GAs accumulation by suppressing the expression of GAs metabolism and signaling-related genes [10,11,24]. In this study, RNA-Seq data found that several GA-related genes, including biosynthesis genes, catabolic genes and signal transduction genes, were downregulated at least at one time point after MC treatment. Unlike previous reports, only few biosynthesis genes, including *CPS*, *KAO1* and *GA3ox* were found to be downregulated at one or more time point, which was not enough to explain why the GAs content decreased significantly. Unlike previous reports, several *GA2**ox* genes were found to show more than 3-fold upregulation after MC treatment in both varieties. Of these genes, 4 were induced 3- to 6-fold by MC at least at one time point in XLZ74, whereas 3 showed more than 7-fold upregulation at 6 dps in SD1068. Additionally, it was found that the content of GA_15_ and GA_4_ decreased in both varieties, with a more significant decrease found in SD1068. GA_12_ is the common precursor for all GAs in higher plants at the final stage of bioactive GA synthesis (Figure 5a) [9]. During the process of biosynthesis of GA_4_, GA_12_ is converted to GA_15_, GA_24_ and GA_9_ in turn, which are catalyzed by two soluble 2-oxoglutarate-dependent dioxygenases (2ODDs) known as GA20ox and GA3ox. Bioactive GAs (GA_1_, GA_3_ and GA_4_) can be transformed into inactive GAs, including GA_34_ and GA_51,_ etc. under the action of GA2ox [10,25]. Catalyzing active GAs into inactive GAs by GA2ox is an important regulatory step in GA metabolism [26]. Up- or downregulation of *GA2**ox* gene can effectively regulate the content of plant endogenous hormones and affect plant growth and development [27,28,29,30,31]. Over-expression of *GA2**ox* gene decreased GA content and resulted in plant dwarfism [32,33,34]. These results prompt us to speculate that the significant reduction in plant height and internode length observed in SD1068 is likely to be a result of the significant MC-induced upregulation of *GA2**ox* genes, leading to significant decrease in GA_4_ content.

### 3.2. MC-Induced Upregulation of Auxin Signaling Genes Is Delayed in SD1068

Auxin plays important roles in the determination of plant growth and development. The inhibited expression of genes related to auxin conjunction, transport and signaling by MC has been previously reported [3,11]. Similarly, our RNA-seq data showed that the expression of several genes related to auxin conjunction (*GH3*) and auxin signaling (*SAUR* and *IAA*) decreased by MC at least at one time point in SD1068, and several *GH3* genes decreased at 1 dps in XLZ74. However, we also found that many auxin-signaling genes (*IAA/AUX* and *SAUR*) were upregulated at 3 and 6 dps in XLZ74 and SD1068, respectively. *AUX/IAA* and *SAUR* have been well recognized for their roles in auxin-mediated responses. The expression of *AUX/IAA* was induced by various plant growth regulator and phytohormones including IAA, 2,4-D, kinetin, 24-epibrassinolide, epibrassinolide and jasmonic acid [35]. Overexpression of *OsIAA1* in rice resulted in reduced plant height [35]. Silencing of *StIAA2* in potato caused increased plant height [35]. Some *SAUR* genes were reported to negatively regulate auxin biosynthesis and transport [36,37,38]. We found a more significant decrease in Trp, IAA-Glc and IAA-Glu content in SD1068 than in XLZ74 at 6 dps (Figure 8), likely due to a more significant increase in the expression of *AUX/IAA* and *SAUR* genes at 6 dps in SD1068 than in XLZ74.

### 3.3. MC-Induced Upregulation of CTK Signaling Genes Is Earlier in SD1068 Than in XLZ74

Cytokinins regulate various aspects of plant growth and development. Unlike GAs and auxin, most CTK-related DEGs were induced by MC, which is consistent with the previous research [11]. Our RNA-seq data showed that the expression of DEGs related to CTK degradation signal pathway increased from 1 dps in MC-treated SD1068, but from 3 dps in MC-treated XLZ74, suggesting a delay of XLZ74 in response to MC compared with SD1068. CTK oxidative decomposition catalyzed by CKX is an important mechanism for regulating the dynamic balance of CTK content [39]. Increasing the content of CTK can rapidly induce the expression of several *CKX* genes [40]. The MC-induced upregulation of *CKX* genes led to increase in the CTK content in both varieties (Figure 8f). CTKs are usually N6-modified adenines such as N6-(δ2-isopentenyl) adenine (iP) and trans-zeatin (tZ), which play important roles in controlling growth and development of plants [41]. Upon MC treatment, SD1068 showed a more significant increase in tZR content than XLZ74 (Figure 8f). A rice *LOG* gene is required to maintain meristem activity and its loss of function results in premature termination of the shoot meristem [21]. We found that several *LOG* genes were downregulated only in MC-treated SD1068, which might partly contribute to reduction in plant height and internode length in SD1068.

### 3.4. MC Treatment Suppresses BR Biosynthesis-Related Genes Only in SD1068

BRs are essential for stem elongation [42]. Previous report found that BR biosynthesis was significantly downregulated by MC treatment [11]. Consistently, our RNA-seq data showed downregulation of several BR biosynthesis genes at 6 dps in SD1068. In contrast, all BR-related DEGs identified in XZ74 were upregulated. Although the mechanism behind the difference is yet to be uncovered, the result is in line with the MC-induced phenotypic outcomes.

### 3.5. MC Treatment Induces Different Profiles of TFs in XLZ74 and SD1068

A number of transcription factors (TFs) were found to be differentially expressed in both varieties following MC treatment, including *MYB*, *WRKY*, *NAC* and *bHLH*, which have been reported to play vital roles in plant growth and metabolism. *Arabidopsis AtMYB91/AS1* was found to be involved in shoot morphogenesis [43]. Wheat *TaGAMYB* expression correlates with internode length and is higher in varieties with longer internode [44]. In apple, mutation in *MdWRKY9* causes dwarfism likely due to direct inhibiting the transcription of genes encoding rate-limiting enzymes involved in BR biosynthesis, leading to reduction in BR production [45]. *Arabidops CUC1* and *CUC2* encoding closely related members of the *NAC* TFs are involved in shoot meristem initiation and specification of shoot organ boundaries [46]. Two maize paralogues, *ZmNAM1* and *ZmNAM2* belonging to the NAC family were associated with shoot apical meristem establishment [47]. Overexpressing a cotton *bHLH* transcription factor, *GhPAS1*, in *Arabidopsis* increases plant biomass, and downregulation of *GhPAS1* in cotton inhibits cotton growth and development, including plant height, fruit branch length and boll size [48]. We found that the expression of most TFs was significantly decreased in MC-treated SD1068 and that less downregulated TF genes were found in XLZ74. The different expression patterns of TF families in the two varieties could be one of the main reasons for their sensitivity to MC treatment.

### 3.6. MC Treatment Suppresses Sugar and Starch Pathway in SD1068

Starch and sucrose are the pivotal components and end products of net photosynthetic rate in different plants such as cotton. Trehalose-6-phosphate phosphatase (TPP) catalyzes the dephosphorylation of trehalose-6-phosphate to trehalose [49]. The changes of trehalose pathway affect plant metabolism and development, including photosynthesis and sucrose utilization [50,51]. Trehalose can regulate the pathway of glucose metabolism, and trehalose metabolism-related gene can regulate the interaction of glucose signal pathway, which can enhance photosynthetic capacity [52,53]. β-amylase is the enzyme that can cleave α-1,4-glycosidic bonds. They act externally on the residues of amylose or amylopectin and produce glucose. The downregulation of sugar and starch-related genes may result in the low glucose and trehalose content in SD1068, resulting in slow growth and reduced plant height in SD1068.

## 4. Materials and Methods

### 4.1. Plant Materials and MC Treatments

Two cotton (*Gossypium hirsutum* L.) varieties Xinluzao 74 (XLZ74) and Shida1068 (SD1068) were used in this study (obtained from Institute of Cotton Research of Chinese Academy of Agricultural Sciences, Anyang, China). Seeds of the two cotton varieties were sown in the experimental field of Cotton Research Institute of Shihezi University (44°27′ N, 85°94′ E, Shihezi, Xinjiang, China).

The concentration 80 mg/L MC is commonly used in commercial production. Except for this concentration level, a lower (40 mg/L) and a higher concentration level (120 mg/L) were set in this study. When the third true leaf fully expanded, seedlings were treated with foliar spraying of 40, 80 or 120 mg/L MC at three time points, respectively. Seedlings sprayed with water were used as controls. The first treatment time point was on 27 May, the second on 4 June, and the third on 13 June 2020. The plant height was measured from cotyledon node to grow point on 10 July, which was 27 days after the 3rd spray. Each treatment at each time point was replicated three times with each replicate consisted of 15 seedlings.

Cotton seedlings treated with 80 mg/L MC (the commonly used concentration) were also used for further measurements, including plant height and the length of the second and third internodes, which were carried out at 0, 2, 4, 6, 8 and 10 days-post-spray (dps). Simultaneously, the MC-treated seedlings were collected for further assays. At 1, 3, and 6 dps, about 1 cm of young stem near the growing point was harvested from seedlings treated with 80 mg/L MC for RNA extraction, transcriptome analysis and endogenous hormone measurements. Control plants from the same time points were also harvested. Each time point had three biological replicates.

### 4.2. RNA Extraction and Sequencing

Total RNA was extracted from the stem segment using a RNAprep Pure Plant Plus Kit (Polysaccharide- and Polyphenolic–rich) (Tiangen Biotech, Beijing, China). RNA integrity was assessed by RNase-free agarose gel electrophoresis and a 2100 bioanalyzer (Agilent Technologies, USA). RNA purity and concentration were determined with a NanoPhotometer sepctrophotometer (Thermo Scientific, California, USA). RNA-Seq libraries were built by using NEBNext^®^ Ultra™ RNA Library Prep Kit for Illumina^®^, and then sequenced on the Illumina Hiseq™ 4000 platform (Illumina, San Diego, CA, USA, 2010) at Novogene Bioinformatics Technology Co., Ltd. (Beijing, China).

### 4.3. RNA-seq Analysis and Identification of Differentially Expressed Genes

After sequencing, 150-bp paired-end reads were generated. Clean reads were obtained by removing low-quality reads (Q ≤ 20), reads containing adapters or with more than 50% N bases. All the clean reads were aligned to the cotton reference genome (*Gossypium hirsutum* (AD1) ‘TM-1’ genome ZJU-improved_v2.1_a1 (https://www.cottongen.org/ (accessed on 2 July 2020)) [54] by using HISAT2 software program [55]. Only reads with a perfect match or one mismatch were retained for further analysis. Gene expression levels were normalized as FPKM (expected number of fragments per kilobase of transcript sequence permillions base pairs sequenced [56]. The DESeq2 was used in pairwise gene expression comparisons [57]. The resulting *p* values were adjusted using the BH [58] approach for controlling the false discovery rate. Genes with a minimum 1.5-fold difference in expression and padj ≤ 0.05 were considered as DEGs. Gene Ontology (GO) and Kyoto Encyclopedia of Genes and Genomes (KEGG) enrichment analysis of the DEGs were implemented by culsterProfiler software. The GO items with adjusted padj value ≤ 0.001 and KEGG pathways with *p* value ≤ 0.001 were considered to be significantly enriched.

### 4.4. qRT-PCR Analysis

Total RNA was extracted from the stem segment as mentioned above. Two μg of total RNA was used for the first-strand cDNA synthesis with the EasyScript One Step gDNA Removal and cDNA Synthesis SuperMix (TransGen Biotech, Beijing, China). Gene specific primers with the product size ranging from 100–250 bp were designed (Appendix A), and *GhUBQ7* (*DQ116441*) was used as the internal control. The qRT-PCR assays were performed in 96-well plates using a LightCycler 480 system (Roche, Basel, Switzerland). According to the manufacturer’s protocol, each PCR reaction contained 5 μL of 2× PerfectStart Green qPCR SuperMix (TransGen Biotech, China), 1 μL of the duluted cDNA template, and 0.2 μL of each primer (10 μM) in a total reaction volume of 10 μL. The PCR reaction conditions were: 1 min at 94 °C and 45 cycles of the thermal cycling of 15 s at 94 °C 15 s at 56 °C and 20 s at 72 °C. All the reactions were measured in triplicate. The relative gene expression level was calculated and normalized using the 2^−^^ΔΔCT^ method [59].

### 4.5. Hormone Quantification

The content of endogenous GAs, IAAs and CTK in the young stem near the growing point was determined using 6 dps samples from both varieties. Plant samples were frozen in liquid nitrogen and ground into powder. Fifty mg of sample powder was weighted and dissolved in 1 mL methanol/water/formic acid (15:4:1, *v*/*v*/*v*). 10 μL internal standard mixed solution (100 ng/mL) was added into the extract as internal standards for the quantication. The mixture was vortexed for 10 min, then centrifuged for 5 min (12,000 r/min, 4 °C). The supernatant was evaporated to dryness and dissolved in 100 μL 80% methanol (*v*/*v*) and filtered with 0.22 μm membrane filter for further LC-MS/MS analysis [60,61]. The extracts were analyzed using an UPLC-ESI-MS/MS system (UPLC, ExionLC™AD, https://sciex.com.cn/ accessed on 5 October 2021); MS, Applied Biosystems 6500 Triple Quadrupole, according to previous reports [62,63,64,65].

### 4.6. Statistical Analyses

Statistical analyses were performed by using Student’s *t*-test. Multiple comparisons were conducted with the SPSS19 software.

## 5. Conclusions

Comparative transcriptome analysis reverals common and unique response of SD1068 and XLZ74 to MC treatment. Generally, MC inhibited GA and BR biosynthesis pathways, and induced auxin and CTK signal transduction pathways, resulting in reduction in GA and IAA content and increase in CTK content. Compared to XLZ74, SD1068 showed a more significant reduction in GA_4_, a delayed upregulation of several *AUX/IAA* and *SAUR* genes that led to more significant decrease in Trp, IAA-Glc and IAA-Glu. Additionally, MC treatment repressed more TFs related to phytohome homeostasis and plant growth and development in SD1068 than in XLZ74. Consequently, upon MC treatment, SD1068 had a severer reduction in plant height and internode length than XLZ74.

## Figures and Tables

**Figure 1 ijms-23-05043-f001:**
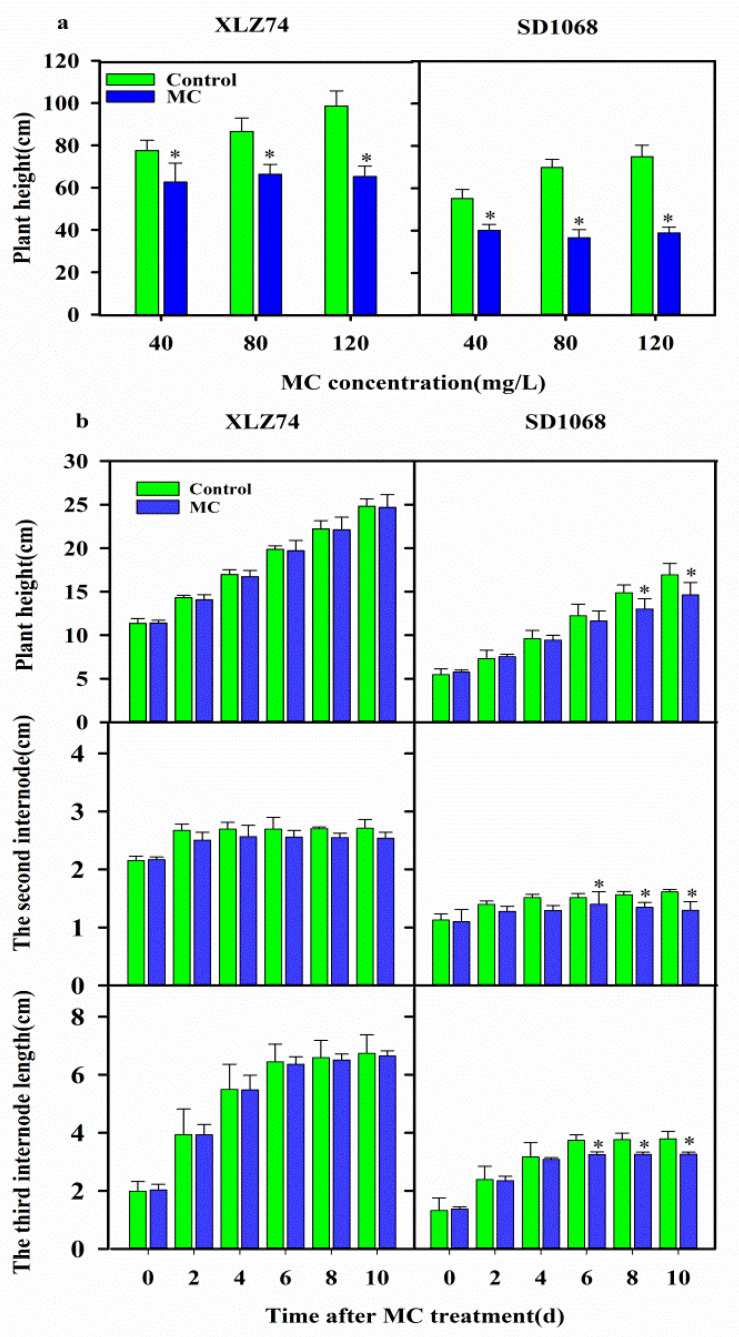
Effects of Mepiquat chloride (MC) treatment on plant height and internode elongation. (**a**) Plant height of cotton seedlings in response to different concentration of MC (40, 80 and 120 mg/L). Each MC concentration was applied to the seedlings three times in a period of 17 days. The plant height was measured 27 days after the 3rd spray. (**b**) Dynamic changes in the plant height and the length of the second and third internodes at 2, 4, 6, 8 and 10 d after spray of 80 mg/L MC. Cotton seedlings with the third true leaf fully expanded were used in the experiments. Error bars represent the SD of three biological replicates, * *p* < 0.05, Student’s *t*-test.

**Figure 2 ijms-23-05043-f002:**
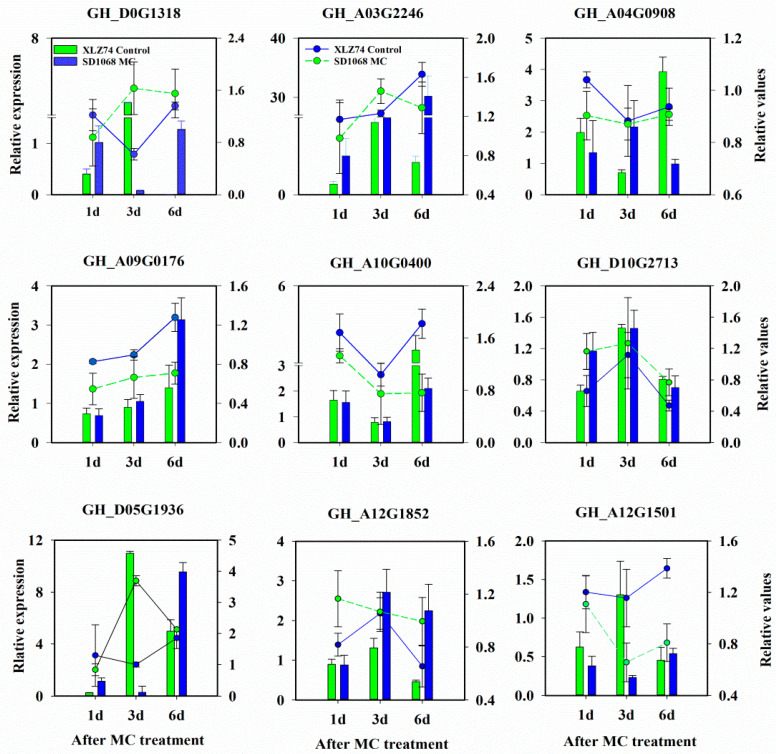
qRT-PCR validation of the RNA-seq results of 9 randomly selected genes. Y-axis represents the relative expression level based on qRT-PCR (displayed as columns) and relative values (the ratio of FPKM between MC and Water treatment, displayed as lines) of the gene shown on top of the graph. Error bars represent the SD of three biological replicates.

**Figure 3 ijms-23-05043-f003:**
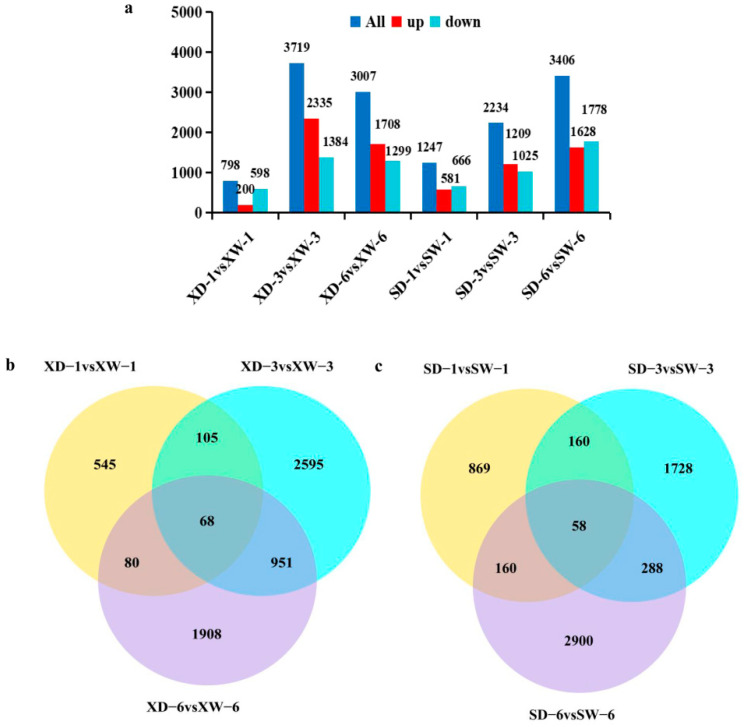
MC-induced DEGs in both cultivars at the three time points. (**a**) The number of DEGs up- or downregulated at 1, 3 and 6 d after MC treatment. X axis represents different comparisons and Y axis represents the number of DEGs identified. (**b**,**c**) Venn diagrams showing the number of DEGs unique to each time point or overlapping between different time points in XLZ74 (**b**) and SD1068 (**c**). XD and XW, XLZ74 treated with MC and water, respectively. SD and SW, SD1068 treated with MC and water, respectively.

**Figure 4 ijms-23-05043-f004:**
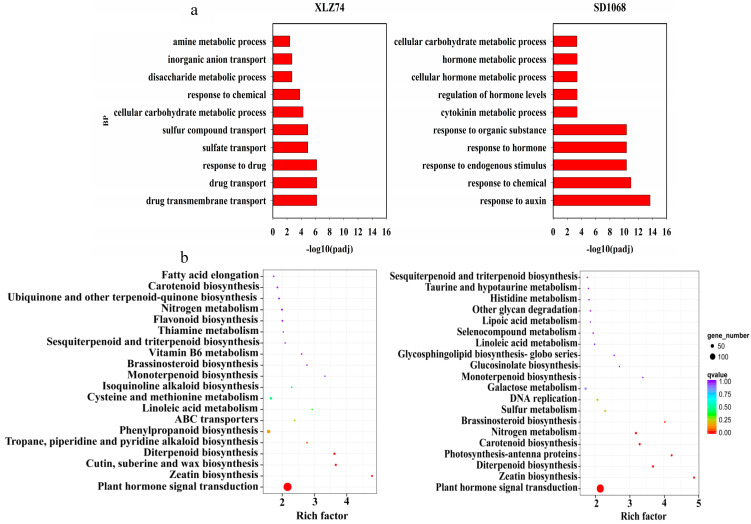
Gene Ontology (GO) enrichment and Kyoto Encyclopedia of Genes and Genomes (KEGG) pathway analyses of DEGs between MC and water treatments in the two varieties. (**a**) Enriched biological process GO terms (padj < 0.001). X-axis shows -log10 (padj) value, and Y-axis shows the top ten enriched GO terms. (**b**) Enriched KEGG pathways. For each variety, all DEGs identified at the three time points were used in functional classification. Y-axis shows the KEGG pathways, and X-axis shows the Rich factor. The size and color of each circle represent the number of enriched genes in the corresponding pathway and the −log10 (*q* value) value, respectively.

**Figure 5 ijms-23-05043-f005:**
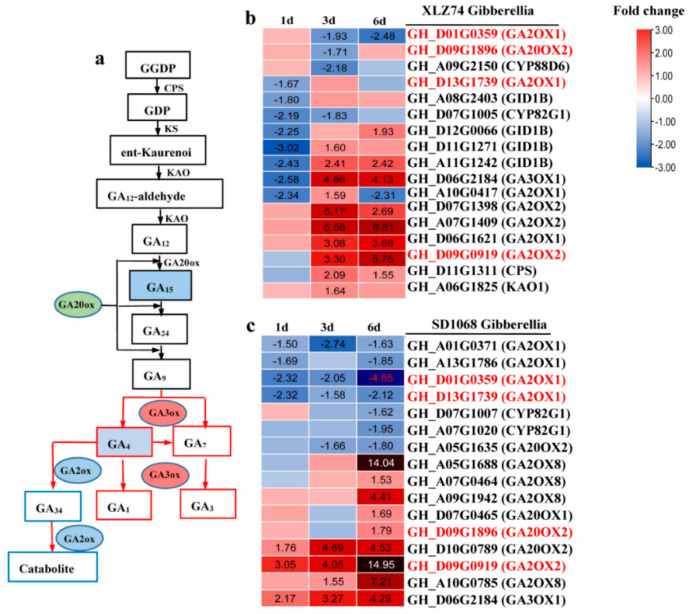
The expression patterns of GA-related genes in XLZ74 and SD1068. The results were based on the RNA-seq data. (**a**) A simple diagram to show the relationship of the genes and GA. (**b**,**c**) The heatmaps were generated based on the RNA-seq data. The number in box indicates the expression fold change of the gene (only those with a ≥1.5-fold difference are shown). MC-induced downregulated and upregulated expression are indicated by blue and red colors, respectively. Gene ID marked with red color represents common gene between XLZ74 and SD1068. All the numbers in heatmap were fold change.

**Figure 6 ijms-23-05043-f006:**
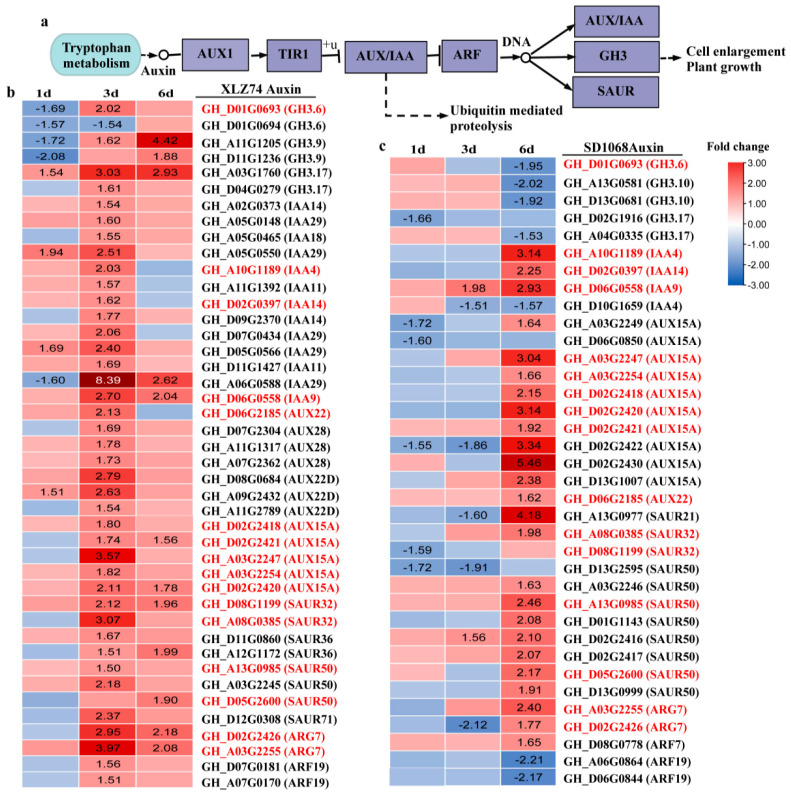
The expression patterns of Auxin-related genes in XLZ74 and SD1068. The results were based on the RNA-seq data. (**a**) A simple diagram to show the relationship of the genes and Auxin. (**b**,**c**) The heatmaps were generated based on the RNA-seq data. The number in box indicates the expression fold change of the gene (only those with a ≥1.5-fold difference are shown). MC-induced downregulated and upregulated expression are indicated by blue and red colors, respectively. Gene ID marked with red represents common gene between XLZ74 and SD1068. All the numbers in heatmap were fold change.

**Figure 7 ijms-23-05043-f007:**
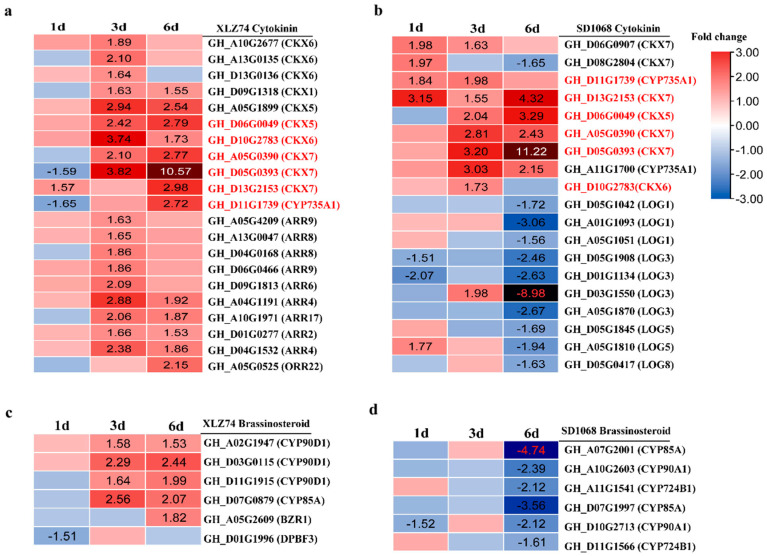
The expression patterns of CTK (**a**,**b**)- and BR (**c**,**d**)-related genes in XLZ74 and SD1068. The results were based on the RNA-seq data. The number in box indicates the expression fold change of the gene (only those with a ≥ 1.5-fold difference are shown). MC-induced downregulated and upregulated expression are indicated by blue and red colors, respectively. Gene ID marked with red represents common gene between XLZ74 and SD1068. All the numbers in heatmap were fold change.

**Figure 8 ijms-23-05043-f008:**
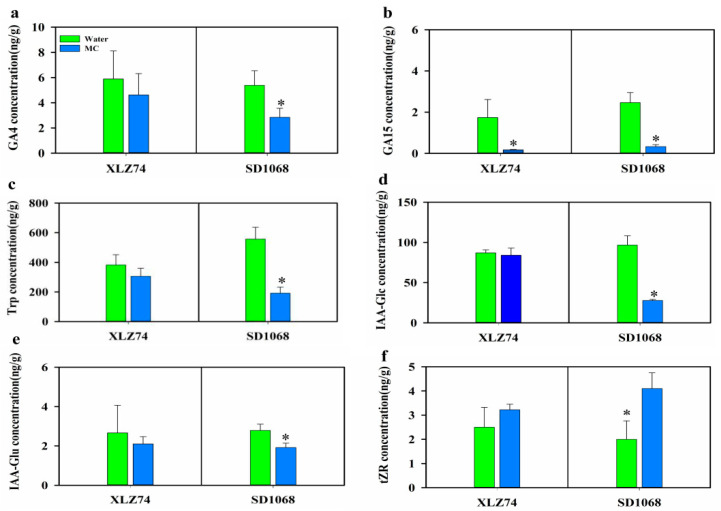
The contents of endogenous hormone in young stem from XLZ74 and SD1068 at 6d after MC treatment. (**a**,**b**) the contents of GA_4_ and GA_15_; (**c**–**e**) the content of Trp, IAA-Glc and IAA-Glu; (**f**) the tZR content. Error bars represent the SD of three biological replicates. * *p* < 0.05, Student’s *t*-test.

**Figure 9 ijms-23-05043-f009:**
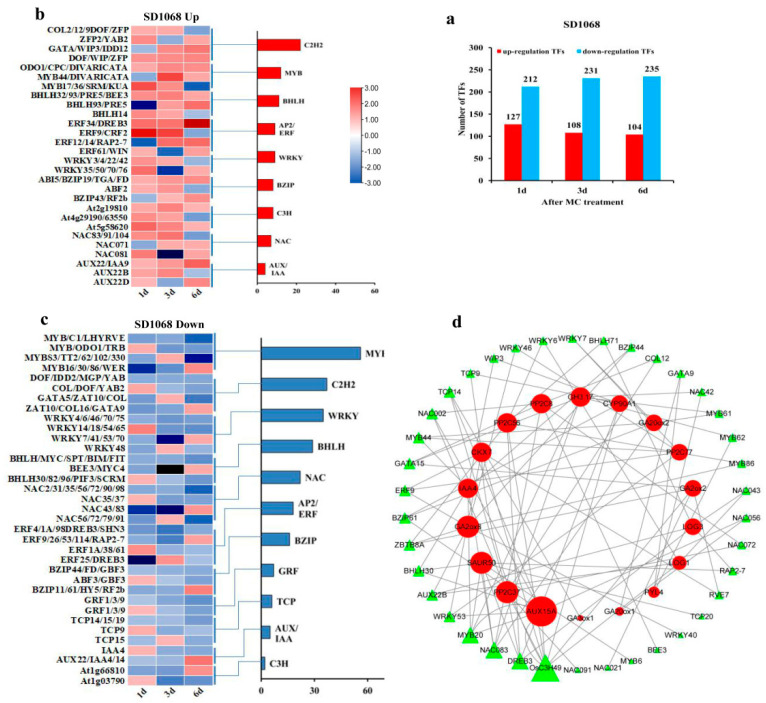
The number and expression patterns of the up- or downregulated transcription factors (TFs) in SD1068. (**a**) The total number of up- or downregulated TFs in SD1068 at three time points after MC treatment. (**b**) The expression pattern of upregulated TFs in SD1068. (**c**) The expression pattern of representative downregulated TFs in SD1068. (**d**) The co-expression network of hormone-related genes and TFs in SD1068. The larger triangles on TFs and circles on hormone-related genes indicated the stronger connectivity, which meant that the corresponding genes were more important for the interaction of TFs with hormone metabolism and signaling genes.

**Figure 10 ijms-23-05043-f010:**
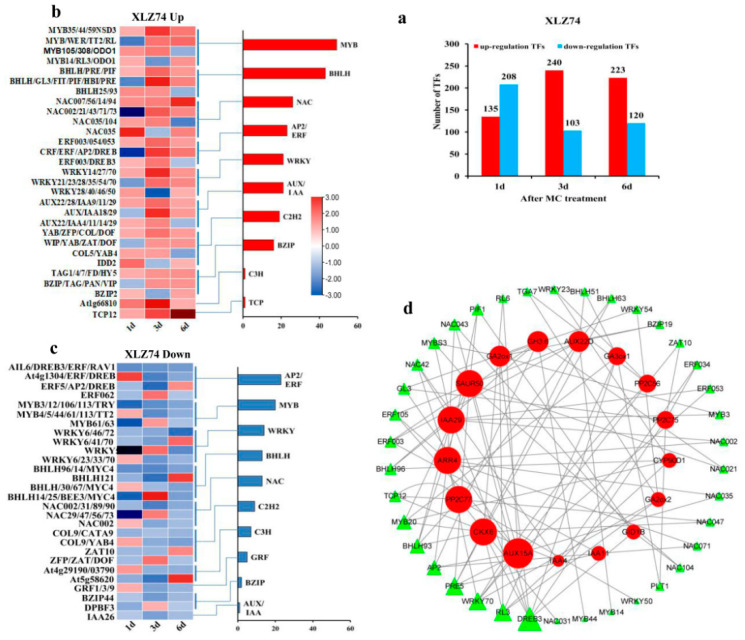
The number and expression pattern of the up- and downregulated transcription factors (TFs) in XLZ74. (**a**) The total number of up- or downregulated TFs in XLZ74 at three time points after MC treatment. (**b**) The expression pattern of representative upregulated TFs in XLZ74. (**c**) The expression pattern of representative downregulated TFs in XLZ74. (**d**) The co-expression network of hormone-related genes and TFs in XLZ74. The larger triangles on TFs and circles on hormone-related genes indicated the stronger connectivity, which meant that the corresponding genes were more important for the interaction of TFs with hormone metabolism and signaling genes.

**Figure 11 ijms-23-05043-f011:**
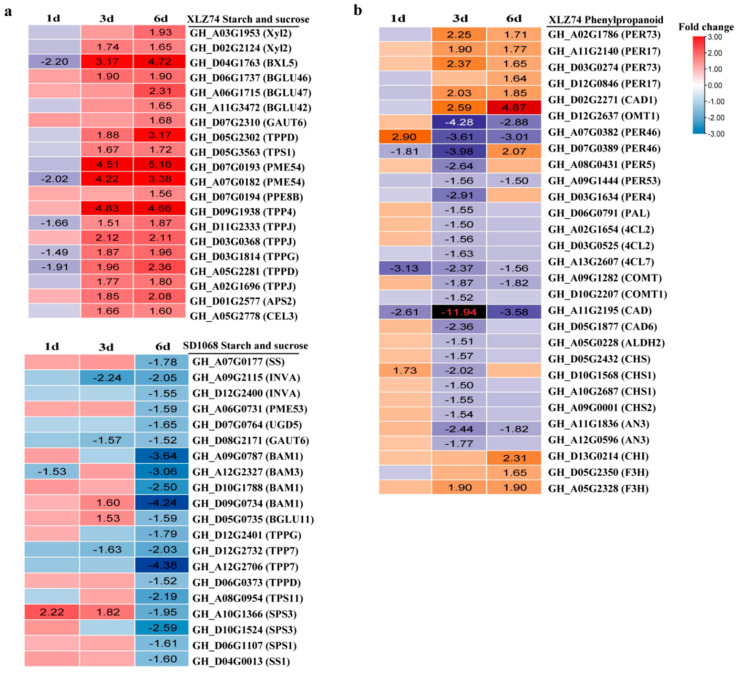
The expression patterns (RNA-seq based data) of starch and sucrose metabolism and phenylpropanoid metabolism-related genes. (**a**) Heatmap showing the expression patterns of the genes related to starch and sucrose metabolism in XLZ74 (top panel) and SD1068 (bottom panel). (**b**) Heatmap showing the expression patterns of the genes related to phenylpropanoid metabolism in XLZ74. The number in box indicates the expression fold change of the gene (only those with a ≥1.5-fold difference are shown). MC-induced downregulated and upregulated expression are indicated by blue and red colors, respectively. All the numbers in heatmap were fold change.

## Data Availability

Raw sequencing data can be accessed through the Gene Expression Omnibus with the accession number PRJNA800117.

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
