# Peer review of "Transcriptome Profiling Provides New Insights into the Molecular Mechanism Underlying the Sensitivity of Cotton Varieties to Mepiquat Chloride"

_ijms, 2022, doi:10.3390/ijms23095043_

Round 1
Reviewer 1 Report
The manuscript gives relevant information about how cotton plants respond to mepiquat chloride, a plant growth retardant. The work has interesting results and is well written. A edited version of the ms has been uploaded with my suggestions.

Author Response
Response to Reviewer 1 Comments
Point 1: The manuscript gives relevant information about how cotton plants respond to mepiquat chloride, a plant growth retardant. The work has interesting results and is well written. A edited version of the ms has been uploaded with my suggestions.
Our response: We are grateful to you for your kind letter of encouragement, along with the constructive comments.
Point 2: Avoid the use of personal pronouns
Our response: Many thanks to reviewer for this comment. The personal pronouns have been avoided in revised manuscript. Please refer to line (16, 18, 142, 235, 461).
Point 3: to understand, Gibberellins, GAs, (IAA)
Our response: We are sorry for these mistakes. We have corrected them. Please refer to line (25, 26, 32).
Point 4: give full name(dps)
Our response: Many thanks to reviewer for this comment. We have added the full name “days-post-spray”. Please refer to line (33).
Point 5: Please change this KW ”growth inhibitor”
Our response: Many thanks for this constructive comment. We have replaced the KW “growth inhibitor” with a new keyword “transcription factor”. Please refer to line (43).
Point 6: Italic,fiber,there are dozens of GAs,ethylene
Our response: We are sorry for these mistakes. We have corrected them. Please refer to line (46, 57, 61).
Point 7: given values line(69,70)
Our response: Many thanks to the reviewer for this comment. According to our experience, MC-sensitive cotton varieties need only 1 to 2 times of foliar sprays 15-30 g/hm2 MC during their growth season, but MC-insensitive one need 3-4 times of spray of 75 g/hm2 MC in order to balance their vegetative and reproductive growth. We have added the MC concentration data and relevant literature support [17] in revised manuscript. Please refer to line (71, 72).
Point 8: Short this title(84)
Our response: Many thanks to the reviewer for this comment. The title has been changed to “The Sensitivity of Cotton Varieties to MC”. Please refer to line (87).
Point 9: what does this mean“Scarcely”(101)
Our response: We are sorry for the unclear statement. The “Scarcely” has been changed to “not significantly”. Please refer to 1ine (104).
Point 10: Auxin is also a hormone there is some redundancy in the different categories.()
Our response: We agreed entirely with this comment. We have detected the go term“response to auxin”. Please refer to line (219).
Point 11:Specify“drug transport”line (176).
Our response: Many thanks to the reviewer for this comment. Drug transport term is related to many gene family, so it’s hard to clearly state. To avoid confusing readers, we deleted the term “drug transport” in revised manuscript. Please refer to line (223).
Point 12:The name of the genes must be in italics, check through the manuscript.
Our response: We are sorry for these mistakes. According to editor’s suggestion, we have checked the manuscript carefully and change the gene name to italics. Some names that have not been changed to italics are enzyme name.
Point 13: Why only these types of conjugated molecules were analysed ,what about free IAA? Please indicate the type of GAs analysised, only GA4? perhaps in the M&M section.
Our response: We are sorry for the unclear statement. All types of IAAs and GAs were determined by LC-MS/MS in this study. Only Trp, IAA-Glu, IAA-Glc , GA4 and GA15 showed significant content changes compared to CK. Free IAA and other type of IAAs and GAs showed no significant content changes. Therefore, only Trp, IAA-Glu, IAA-Glc, IAA-Asp, GA4 and GA15 were described in our study. In order to help readers to understand this section, we have stated why only these types of IAAs and GAs in our results. Please refer to line (329-335).
Point 14: Subscript, check through the text, add a ref
Our response: Many thanks for this comment. We have checked the manuscript carefully and change the number to subscript and we have added a ref [9]. Please refer to line (468).
Point 15: These compounds are not phytohormones. line (438)
Our response: Many thanks for this comment. We are sorry for the mistake. We have changed “phytohormones” to “plant growth regulator and phytohormones ”. Please refer to line (489).
Point 16: Please change this sentence. line (438)
Our response: Many thanks for this comment. We have change this sentence to “The plant height was measured from cotyledon node to grow point on July 10th, which was 27 days afer the 3rd spray.” Please refer to line (568-569).
Point 17: The cientific name of the species must be in italics, check through the reference
Our response: Many thanks for this comment. We have checked through the references and corrected them.
Point 18: Uniformize the way references are written here there is a after sci . In the previous ref no
Our response: Many thanks for this comment. We have corrected them.

Reviewer 2 Report
Thanks for inviting to review this article. I have inserted my detail comments in the pdf file. See attached pdf.
Line 68: Lack of data support
Line 86: and
Lines 88-89: Any pre experiment to determine these three concentration levels?
Line 90: Please homogenize the two sets of data to show the arguments more intuitively
Line 93: Why did the plant height of SD1068 increase at 2 dps, while XLZ74 decreased at 2 dps? Any molecular traits display?
Line 97: Any data on the difference between the second and third internode of MC transport ?
Lines 101-103: The rate of decreasing of plant height and second internode in XLZ74 looks different from third internode in XLZ74 and the three traits in SD1086, any molecular proof?
Line 196: The picture resolution doesn't seem to meet the requirements
Line 287: Many hormones promote growth at low concentrations and inhibit growth at high concentrations, why did you choose this time point? Any pre-treatment data support it?
Lines 297-298: This statement is inconsistent with line 287
Lines 408-409: Up-regulation of GA2ox genes result in the decreasing of GA content, it seems these genes are not GA biosynthesis genes, they are the resolved genes or inactive genes? While which genes are the GA biosynthesis genes? How did you identify these genes from the RNA-seq data?
If these up-regulated GA2ox genes were GA biosynthesis, they lead to a too high GA content to inhibit the plant height, why the GA content you displayed in Figure 8a and b decreased?
All the numbers in your heatmap (figure 5,6,7,8), they are Fold Change or Log2 Fold Change? Please mark clearly in the heatmap and text.
Line 426: Your proof can not support this point. At 1 and 3 dps in SD1068, more genes are down-regulated in figure 6b and c, which lead to a decrease of IAA in figure 8c,d,e.
As a compensation mechanism, these genes are up-regulated at 6dps, but you only have data at 6dps in figure 8, need more data at 1 and 3 dps to support it.
Lines 434-437: Two opposite sentence
Line 506: Please provide the correct statement
Line 511: An incomplete sentence?
Lines 521-523: These two sentences conflict with the statement in highlighted line 509.
Lines 554-556: No gel electrophoresis and melting curve displayed.
Lines 562-563: Please cite references.
Line 568: Use Arabic numerals correctly

Author Response
Response to Reviewer 2 Comments
Point 1: Line 68: Lack of data support
Our response: Many thanks to the reviewer for this comment. According to our experience, MC-sensitive cotton varieties need only 1 to 2 times of foliar sprays 15-30 g/hm2 MC during their growth season, but MC-insensitive one need 3-4 times of spray of 75 g/hm2 MC in order to balance their vegetative and reproductive growth. We have added the MC concentration data and relevant literature support [17] in revised manuscript. Please refer to line (71, 72).
Point 2: Line 86: and
Our response: We are sorry for the mistake. We have corrected it. Please refer to Line 89.
Point 3: Lines 88-89: Any pre experiment to determine these three concentration levels?
Our response: This is indeed a good question. We are sorry that we didn’t do pre experiment to determine these three concentration levels. The reasons for choosing these three concentrations were as follows. The concentration 80 mg/L is commonly used in commercial production. Based on this concentration, we also set a lower concentration (40 mg/L) and a higher concentration (120 mg/L) to identify the sensitivity of two varieties. Under these three concentration treatments, the plant height of SD1068 degraded more than that of XLZ74. Finally, the commonly used concentration 80 mg/L was selected for subsequent experiments. In order to make readers better understand how we set these concentrations, we briefly describe the above idea which were added in the materials and methods section. Please refer to Line 562-564.
Point 4: Line 90: Please homogenize the two sets of data to show the arguments more intuitively.
Our response: This is a constructive suggestion. We have homogenized the two sets of data by adjusting the y axis to the same. Please refer to Figure 1a.
Point 5: Line 93: Why did the plant height of SD1068 increase at 2 dps, while XLZ74 decreased at 2 dps? Any molecular traits display?
Our response: Many thanks to the reviewer for the careful reading and discovery. Although the plant height of SD1068 increase at 2 dps, XLZ74 decreased at 2 dps, but there was no significant change in plant height for the two varieties according to our observation (Figure 1b). The MC-sensitive variety SD1068 showed significant change in plant height from 8dps (Figure 1b). The plant height increase in SD1068 2 dps may be caused by measurement error, therefore, we are sorry for no molecular traits displaying for the result.
Point 6: Line 97: Any data on the difference between the second and third internode of MC transport?
Our response: Many thanks to the reviewer for this question. In this study, we found that the length of the second internode in SD1068 was reduced by 8.59, 14.54, 7.49, 13.75 and 19.63%, and the third internode by 1.69, 2.70, 13.01, 13.53 and 14.06% at 2, 4, 6, 8 and 10 dps, respctively, indicating that the length of the second internode was inhibited more severely than that of the third internode after MC treatment. We only focused on the length change of the second and third internode after MC treatment, and we are sorry for our negligence of the data on the difference between the second and third internode of MC transport. This question provides us with good ideas, and we will consider relevant research in the future.
Point 7: Lines 101-103: The rate of decreasing of plant height and second internode in XLZ74 looks different from third internode in XLZ74 and the three traits in SD1068,any molecular proof?
Our response: Many thanks to the reviewer for this question. We only conducted the RNA-seq for the young stems, and didn’t conduct the RNA-seq for the second and third internodes. Therefore, we only have molecular evidence of plant height difference between two cotton varieties. Through the RNA-seq of stem, we found that Several GA catabolic genes, GA2ox, were highly induced by MC in both varieties especially in SD1068, consistent with a more significant decrease of GA4 in SD1068. Several AUX/IAA and SAUR genes and CKX genes were induced by MC in both varieties, but with a more profound effect observed in SD1068 that showed a significant reduction of indole-3-acetic acid (IAA) and a significant increase of cytokinin (CTK) at 6 days-post-spray (dps). BR biosynthesis related genes were down-regulated in SD1068, but not in XLZ74. These molecular proof may explain the reasons why SD1068 degraded more in height compared to XLZ74. Please refer to our abstract, Line 28-34. This question provides us with good ideas, and we will consider RNA-seq for internodes in the future.
Point 8: Line 196: The picture resolution doesn't seem to meet the requirements.
Our response: We agree entirely with this comment. We have improved the picture resolution. Please refer to Figure 4.
Point 9: Line 287: Many hormones promote growth at low concentrations and inhibit growth at high concentrations, why did you choose this time point? Any pre-treatment data support it?
Our response: Many thanks to the reviewer for this question. In our experiment, RNA-Seq was conducted at 1, 3 and 6 days after MC treatment. In order to save costs, we did not intend to measure hormone content at all time points. In fact, we firstly used 3 dps for hormone measurement, but we found that the changes of gene expression at 6 dps was not easily related to the changes of hormone content. In order to associate the changes of gene expression with the changes of hormone content, samples at 6 dps was then used for hormone measurement according to previously report (Wang et al., 2014). Since the data of 3 dps and 6 dps were not measured at the same time, the difference between them is very large and difficult to compare, so we only used the data of 6 days. We are sorry for that.
Point 10: Lines 297-298: This statement is inconsistent with line 287
Our response: We are sorry for this mistake. We have deleted the “3d” which is inconsistent with line 287. Please refer to Line 344.
Point 11: Lines 408-409: Up-regulation of GA2ox genes result in the decreasing of GA content, it seems these genes are not GA biosynthesis genes, they are the resolved genes or inactive genes? While which genes are the GA biosynthesis genes? How did you identify these genes from the RNA-seq data?
Our response: Many thanks to the reviewer for this question. We have pointed that GA2ox genes are catabolic genes, which deactivates active GAs in results section, please refer to Line 249, 251, 253. As descried in our discussion section, under the action of GA2ox, bioactive GAs (GA1, GA3 and GA4) can be transformed into inactive GAs, including GA34 and GA51 etc. Please refer to Line 470-472.
Previous report found that the expression of many GA biosynthetic genes were downregulated after MC treatment (Wang et al., 2014). However, only few GA biosynthetic genes identified by KEGG pathway enrichment, including CPS, KAO1 and GA3ox were found to be down-regulated at one or more time point, which was not enough to explain why the GA content decreased significantly. In order to help readers to find GA biosynthesis genes, we added above description in discussion section. Please refer to Line 459-461.
Unlike previous report, several GA degrading GA2ox genes were showed more than 3-fold up-regulation after MC treatment in both varieties. Of these genes, 4 were induced 3 to 6-folds by MC at least at one time point in XLZ74, while 3 showed more than 7-fold up-regulation at 6 dps in SD1068. The high-fold up-regulated GA2ox genes can help us explain why the GA content decreased significantly in cotton stem. Therefore, we have paid more attention to the GA degrading GA2ox genes and gave them more descriptions, and gave less descriptions for GA biosynthetic genes.
Point 12: If these up-regulated GA2ox genes were GA biosynthesis, they lead to a too high GA content to inhibit the plant height, why the GA content you displayed in Figure 8a and b decreased?
Our response: Many thanks to the reviewer for this question. In fact, the GA2ox genes are GAs degrading gene, resulting in the decreasing of GA content, so the GA content we displayed in Figure 8a and b decreased. We have pointed that they are catabolic genes in results section, please refer to Line 249, 251, 253. Simultaneously, we have described the function of GA2ox genes in reported studies in discussion section, and speculated that the significant reduction in plant height and internodes length observed in SD1068 is likely to be a result of the significant MC-induced up-regulation of GA2ox genes, leading to significant decrease of GA4 content. Please refer to Line 471-479.
Point 13: All the numbers in your heatmap (figure 5,6,7,8), they are Fold Change or Log2 Fold Change? Please mark clearly in the heatmap and text.
Our response: We are sorry for the unclear mark. All the numbers in our heatmap (figure 5,6,7,8) are Fold Change. We have marked clearly in the heatmap and text. Please refer to Figure 5, 6, 7, 8 and Line 271, 288, 305-306, 430.
Point 14: Line 426: Your proof can not support this point. At 1 and 3 dps in SD1068, more genes are down-regulated in figure 6b and c, which lead to a decrease of IAA in figure 8c,d,e. As a compensation mechanism, these genes are up-regulated at 6dps, but you only have data at 6dps in figure 8, need more data at 1 and 3 dps to support it.
Our response: This is indeed a constructive suggestion. In order to save costs, we only used samples at 6 dps for hormone measurement according to previously report (Wang et al., 2014), which result in imperfect result. But we are very sorry that we do not have the ability to add more data at 1 and 3 dps. The reasons are as follows.
- We finished the experiment in about 2020. The samples at 1 and 3 dps was not preserved now.
- If we do the experiment again to obtained samples at 1 and 3 dps, which are inconsistent with the samples for RNA-seq in 2020. The plant growth environment has changed in 2022, which may result in a different results with samples in 2020.
- Even if we keep the samples for 1 and 3 days. Since the samples of 1, 3 dps and 6 dps were not measured at the same time, the difference between them is still very large and difficult to compare.
Many thanks to the reviewer for this comment. We must pay attention to the problem in future research. This is because it’s really a good and helpful suggestion.
Point 15: Lines 434-437: Two opposite sentence
Our response: Many thanks to the reviewer for this comment. To avoid ambiguity, we have been deleted the later sentence. Please refer to Line 488-489.
Point 16: Line 506: Please provide the correct statement
Our response: Many thanks to the reviewer for this comment. We have deleted “Province”. Please refer to Line 561.
Point 17: Line 511: An incomplete sentence?
Our response: We are sorry for this mistake. We have changed it to “The plant height was measured from cotyledon node to grow point on July 10th., which was 27 days after the 3rd spray.” Please refer to Line 568-569.
Point 18: Lines 521-523: These two sentences conflict with the statement in highlighted line 509.
Our response: We are sorry for the unclear statement. When the third true leaf fully expanded, seedlings were treated with foliar spraying of 40, 80 or 120 mg/L MC at three time points, respectively. Seedlings sprayed with water were used as controls. The first treatment time point was on May 27th, the second on June 4th, and the third on June 13th, 2020. The plant height was measured from cotyledon node to grow point on July 10th., which was 27 days after the 3rd spray. Each treatment at each time point was replicated three times with each replicate consisted of 15 seedlings. We have corrected it. Please refer to Line 564-570.
Point 19: Lines 554-556: No gel electrophoresis and melting curve displayed.
Our response: We are sorry for our negligence. Because no gel electrophoresis and melting curve displayed, so we have deleted the sentence. Please refer to Line 609.
Point 20: Lines 562-563: Please cite references.
Our response: Many thanks to the reviewer for this comment. We have added a reference [62]. Please refer to Line 617.
Point 21: Line 568: Use Arabic numerals correctly
Our response: Many thanks to the reviewer for this comment. We have corrected it. Please refer to Line 622.

Reviewer 3 Report
Dear Authours,
This manuscript presents clear objective and results, however the results are not shown to be applicable in any way to manipulate the application of Mepiquat chloride (MC). Manuscript only shows transcriptomics results for two cotton varieties with contrasting response to Mepiquat chloride (MC) and differential expressions of various plant growth hormones, however, without discussing the importance of these results or how can breeders or researchers use this information to improve MC sensitivity in majority of commercial cotton cultivars. Moreover, overall work presented in the manuscript is not upto the standards of IJMS. Therefore, in my opinion this manuscript can not be accepted for publication in IJMS.
Author Response
Response to Reviewer 3 Comments
This manuscript presents clear objective and results, however the results are not shown to be applicable in any way to manipulate the application of Mepiquat chloride (MC). Manuscript only shows transcriptomics results for two cotton varieties with contrasting response to Mepiquat chloride (MC) and differential expressions of various plant growth hormones, however, without discussing the importance of these results or how can breeders or researchers use this information to improve MC sensitivity in majority of commercial cotton cultivars. Moreover, overall work presented in the manuscript is not upto the standards of IJMS. Therefore, in my opinion this manuscript can not be accepted for publication in IJMS.
Our Response: We have read the referee’s comments very carefully ,we found the referee’s comments most helpful, which provided a good idea for our further research. The referee’s comments has been rejected this paper that it lacked application of mepiquat chloride data. We think different referees may have different opinions on the goal of this article, or focus is different. In our experiment, the goal of our research focus on molecular mechanism underlying the sensitivity of cotton varieties to mepiquat chloride. In addition, about mepiquat chloride application, there have large amounts literature reports, which can provide useful informations for mepiquat chloride application. How to improve MC sensitivity in majority of commercial cotton cultivars, which is an important problem in cotton production. This article do not give a clear solution, but should be probed into for further study. We greatly appreciate your help and that of the referees concerning improvement to this paper. I hope that the revised manuscript is now suitable for publication.

Round 2
Reviewer 3 Report
Author's revisions did not answer the major issue with the manuscript or applicability of the results of this research. Any revision in the manuscript is still unable to provide the concrete information or suggestions that can be applied to utilize the results of this research to either improve applicability, sensitivity of Mepiquat chloride (MC), for enhanced efficiency in controlling vegetative overgrowth or applicability of this chemical to larger group of cotton varieties than it is currently. In conclusion, in my opinion, this research does not have the standard of impact that it can be accepted to be published in IJMS.